# Asphericity of Somatostatin Receptor Expression in Neuroendocrine Tumors: An Innovative Predictor of Outcome in Everolimus Treatment?

**DOI:** 10.3390/diagnostics10090732

**Published:** 2020-09-22

**Authors:** Christoph Wetz, Julian Rogasch, Philipp Genseke, Imke Schatka, Christian Furth, Michael Kreissl, Henning Jann, Marino Venerito, Holger Amthauer

**Affiliations:** 1Department of Nuclear Medicine, Charité-Universitätsmedizin Berlin, corporate member of Freie Universität Berlin, Humboldt-Universität zu Berlin, and Berlin Institute of Health, Augustenburger Platz 1, 13353 Berlin, Germany; julian.rogasch@charite.de (J.R.); imke.schatka@charite.de (I.S.); christian.furth@charite.de (C.F.); holger.amthauer@charite.de (H.A.); 2Division of Nuclear Medicine, Department of Radiology and Nuclear Medicine, University Hospital Magdeburg A.ö.R., Otto-von-Guericke University Magdeburg, Leipziger Strasse 44, 39120 Magdeburg, Germany; philipp.genseke@med.ovgu.de (P.G.); michael.kreissl@med.ovgu.de (M.K.); 3Department of Hepatology and Gastroenterology, Charité-Universitätsmedizin Berlin, corporate member of Freie Universität Berlin, Humboldt-Universität zu Berlin, and Berlin Institute of Health, Augustenburger Platz 1, 13353 Berlin, Germany; henning.jann@charite.de; 4Department of Gastroenterology, Hepatology and Infectious Diseases, University Hospital Magdeburg A.ö.R., Otto-von-Guericke University Magdeburg, Leipziger Strasse 44, 39120 Magdeburg, Germany; marino.venerito@med.ovgu.de

**Keywords:** neuroendocrine tumors, asphericity, everolimus

## Abstract

Background: in patients with gastroenteropancreatic neuroendocrine tumors (GEP-NET), the mTOR inhibitor everolimus is associated with significant improvement in progression-free survival (PFS). This study evaluated the lesional asphericity (ASP) in pretherapeutic somatostatin receptor (SSR) imaging as the first imaging-based prognostic marker for PFS. Methods: this retrospective bicentric cohort study included 30 patients (f = 13, median age, 66.5 (48–81) years) with pretherapeutic [^111^In-DTPA^0^]octreotide scintigraphy (Octreoscan^®^). ASP of functional volumes of up to three leading lesions per patient (*n* = 74) was calculated after semiautomatic, background-adapted segmentation. Uni- and multivariable Cox regression regarding PFS for clinical factors and the maximum ASP per patient was obtained. Results: all 30 patients showed metachronous or progressive liver metastases. ASP, primary tumor site, metastases pattern, and prior peptide receptor radionuclide therapy (PRRT) were significantly associated with PFS in univariable Cox regression. Only ASP > 12.9% (hazard ratio (HR), 3.33; *p* = 0.024) and prior PRRT (HR, 0.35; *p* = 0.043) remained significant in multivariable Cox. Median PFS was 6.7 months for ASP > 12.9% (95% confidence interval (CI), 2.1–11.4 months) versus 14.4 (12.5–16.3) months for ASP ≤ 12.9% (log-rank, *p* = 0.028). Conclusion: pretherapeutic ASP of SSR positive lesions independently predicted PFS for treatment with everolimus in GEP-NET. ASP may supplement risk-benefit assessment before patient inclusion to treatment.

## 1. Introduction

Neuroendocrine tumors (NET) of the bronchopulmonary (lung-) or gastroenteropancreatic system (GEP-NET) are classified as a rare and heterogeneous tumor entity with substantially rising incidence over the last two decades [1,2]. Due to the mostly slow-growing nature of these tumors and the lack of early symptoms, NET are frequently metastasized, and systemic treatment is required. In addition to somatostatin analogues, chemotherapeutic regimes, and radiolabeled peptide receptor treatments (e.g., PRRT), tumor-targeted therapies such as tyrosine kinase inhibitor (TKI, i.e., sunitinib) and especially mammalian target of rapamycin (mTOR) inhibitors (everolimus) are well established [3,4,5,6]. In particular, three large phase III placebo-controlled randomized trials showed the efficacy of everolimus in patients with well-differentiated lung- and gastrointestinal (GI)-NET (RADIANT-2 and RADIANT-4) [3,5] as well as in patients with advanced pancreatic tumors (P)-NET (RADIANT-1 and RADIANT-3) [4,7]. In addition, RADIANT trials demonstrated an improved progression-free survival (PFS) in patients with advanced NET when combining everolimus with somatostatin analogs (SSA), regardless of previous SSA exposure [3,6,7].

Based on these results, the European Neuroendocrine Tumor Society (ENETs) guidelines recommend everolimus as a second-line therapy for advanced P-NEN and as a third-line therapy for GI-NEN [8].

However, due to biological heterogeneity of NET, identification of patients who will respond to treatment remains challenging. As a result, there is an unmet clinical demand to identify patients who are not likely to benefit and thus, avoid the expense and inconvenience of unnecessary therapy. Therefore, prognostic or predictive markers to identify patients that are likely to benefit from everolimus are needed.

Somatostatin receptor (SSR) -specific imaging with scintigraphy (e.g., [^111^In]In-octreotide scintigraphy) or positron emission tomography (PET) is nowadays routinely assessed at initial diagnosis and during follow-up. Despite its use as disease-specific functional imaging, no image-derived parameters are established as prognostic or predictive factors in NET. So far, the asphericity (ASP), a measure of shape irregularity of the functional lesion volume in SSR imaging, has shown prognostic value in NET patients undergoing PRRT [9,10].

This study examined if ASP of NET lesions in SSR imaging before initiation of everolimus treatment could predict PFS.

## 2. Methods

### 2.1. Patients

In a retrospective bicentric cohort study, a total of thirty patients (male, *n* = 17; female, *n* = 13; median age, 66.5 (48–81) years) with histologically proven GEP-NEN were analyzed. All patients had progressive, metastasized NET (Ki–67 range between 1 and 25% with a median of 5%), and curatively intended therapy was not a treatment option in any of the analyzed patients. Further inclusion criteria were as follows: (1) decision on everolimus was determined by interdisciplinary consensus, (2) SSR-positive lesions determined by [^111^In]In-octreotide scintigraphy (OctreoScan^®^) with single-photon emission computed tomography (SPECT) prior to therapy, (3) metastases could be clearly delineated on SPECT imaging, (4) the patients underwent follow-up imaging by contrast enhanced (CE)-CT/MRI. The association between ASP and morphological treatment response was determined using baseline pretherapeutic CT/MRI (mean = 3 weeks, range: 1–7 weeks prior to everolimus) and follow-up imaging every 3–6 months.

The study protocol was approved by the Otto-von-Guericke-University (reference ID: RAD318; vote, 26 October 2016) and Charité-Universitätsmedizin Berlin Institutional Committee (ID: EA2/006/20; vote 28 January 2020) in accordance with the ethical standards of the institutional research committees and with the 1964 Helsinki declaration and its later amendments or comparable ethical standards. All patients gave written and informed consent to the procedures for scientific analysis of the obtained data.

### 2.2. Everolimus

All patients were treated with everolimus (Afinitor^®^, Novartis Europharm Limited, Dublin, Ireland) and underwent treatment in concordance with the RADIANT-3 treatment protocol [4]. Briefly, everolimus inhibits mTOR, a serine–threonine kinase that stimulates growth factors, cell proliferation and activates angiogenesis [11]. Patients underwent therapy at a dose of 10 mg once daily, and treatment was continued until morphological progression of disease. In the event of treatment-related toxic side effects, which led to any dosage reduction of everolimus, patients were excluded from the current analysis.

### 2.3. SPECT/CT Imaging

SSR-positivity was verified by [^111^In-DTPA^0^]octreotide scintigraphy (OctreoScan) using a dual head general purpose camera, either a Discovery NM/CT670 SPECT/CT (GE, Haifa, Israel) or Symbia T6 SPECT/CT (Siemens Healthineers, Erlangen, Germany) according to standard protocols.

Only lesions with at least a Krenning score [12] of 3—a score for visual assessment and stratification of the SSR-expression—were considered for further evaluation. Planar imaging and SPECT with integrated low-dose CT (X-ray tube current = 40 mAs) were performed after intravenous administration of approximately 6 mCi (median MBq, 222 [range, 181.3–265.5]) of OctreoScan^®^ (Mallinckrodt Medical, Petten, The Netherlands). Additionally, using the Discovery NM/CT 670 camera, SPECT with low-dose CT of the thorax and abdomen was performed at 4 and 24 h post-injection (p.i.) (SPECT-ldCT4 and SPECT-ldCT24) using MEGP collimators and energy window settings (60 frames; 40 s per frame (step and shoot); 6° angles; matrix, 128 × 128; bed position overlap, approx. 4 cm; Epeak = 208 keV ± 10%). SPECT data were reconstructed iteratively (ordered subset expectation maximization (OSEM); 2 iterations, 10 subsets; voxel size, 4.4 × 4.4 × 4.4 mm^3^)) with CT-based attenuation correction using the low-dose CT (maximum tube current, 40 mA, 120 kV, 3.75 mm slice thickness) as previously described [9]. Using the Symbia T6 camera, SPECT/low-dose CT of the thorax and abdomen was performed 24 h p.i. using a MEGP collimator (64 steps (128 views); 15 s per step (step and shoot), arc per detector, 180°, approx. 3° per step; matrix, 128 × 128; Epeak, 208 keV ± 10%). SPECT data was reconstructed iteratively (OSEM Flash 3D; 8 iterations, 4 subsets; voxel size, 4.8 × 4.8 × 4.8 mm^3^) with CT-based attenuation correction (low-dose CT).

In addition, response assessment was evaluated with a dedicated workstation (Xeleris workstation, GE Healthcare, Waukesha, WI, USA) or INFINITT (INFINITT Healthcare Co., Ltd., Seoul, Korea) at standard clinical settings. 

### 2.4. Image Analysis

Pretherapeutic SPECT data was used to derive ASP of individual lesions for assessment of shape irregularity of the SSR-positive lesion volume as previously described using dedicated software (ROVER version 2.1.20, ABX, Radeberg, Germany) [9]. Briefly, the volumes of interest (VOI) of hepatic lesions were delineated using a background-adapted threshold-based algorithm relative to the maximum activity in the lesion [13]. ASP was calculated as the relative deviation of the lesion volume’s surface from the surface of an isovolumetric perfect sphere [14]. Figure 1 illustrates orthogonal slices of two representative examples.

For lesion-based analysis, a VOI was delineated separately for each liver metastasis. If the volumes of two neighboring metastases could not be separated on SPECT images, these metastases were excluded from analysis. In the case of more than three liver metastases, the three lesions with the most representative, strongest SSR uptake on visual assessment were delineated. With the aim of achieving the best reproducibility of the results, we concentrated only on liver lesions and did not assess small lesions (<2.5 mL). Lesions that had been previously treated with microwave or radiofrequency ablation were excluded from the analysis.

Imaging-based response evaluation was performed in accordance with the RECIST 1.1 criteria [15]. Definition of VOIs and calculation of the diameter were performed in consensus by two physicians. Morphological assessment was generally performed by contrast-enhanced (CE)-CT. If available, CE-MRI was used instead (*n* = 19/30; 63%). Lesions with a short axis of <10 mm were excluded from response evaluation in accordance with the RECIST 1.1 guidelines [15].

### 2.5. Statistical Analysis

Data analyses were accomplished using SPSS 22 (IBM Corporation, Armonk, NY, USA). Descriptive values were expressed as median, interquartile range (IQR), and range, unless otherwise specified. ASP values were compared between groups of patients with high vs. low ASP using the Mann–Whitney U test. PFS was calculated from initiation of treatment with everolimus to the date of progressive disease according the RECIST 1.1. criteria or death. Kaplan–Meier (KM) analysis and log-rank test as well as univariable Cox regression with respect to PFS were performed for the maximum ASP measured per patient. Multivariable Cox regression was performed including significant parameters in univariable Cos regression. Concordance of the variables “presence of bone metastases” and “number of metastatic sites ≥2” was examined with McNemar’s test. ASP values were binarized into high vs. low ASP using the optimal cut-off based on the minimal *p* value in log-rank test using the Charité cut-off finder [16]. Statistical significance was assumed at a *p* value <0.05.

## 3. Results

### 3.1. Patient and Tumor Characteristics

Seventy-eight patients with metastatic and progressive GEP-NETs underwent second- to third-line treatment (median of two prior systemic therapies; range, 1–4) with everolimus between 2008 and 2019 and 30/78 (38.5%) fulfilled inclusion criteria. In particular, the reasons for study exclusion were dosage reduction (*n* = 28), treatment with everolimus was interrupted or even stopped (*n* = 16), and loss to follow-up (*n* = 4). In 17/30 (56.7%) patients, the primary occurred in the pancreas; in 9/30 (30%), the primary was located in the gastrointestinal tract and 4/30 (13.3%) patients suffered from cancer of unknown primary (CUP). 16/30 patients did not undergo surgical resection because of locally advanced disease at diagnosis. Among the 30 patients, 8 had undergone liver-directed treatment including debulking surgery (*n* = 3) or minimal invasive techniques (*n* = 5; selective internal radiation therapy, *n* = 2; radiofrequency ablation, *n* = 2; or microwave ablation, *n* = 1). In all patients, everolimus was initiated for metachronous or progressive liver metastases. The proliferation index Ki-67 (median, 5%; range, 1 to 25%) was obtained from either the primary tumor (*n* = 16), from at least one representative metastasis (*n* = 8), or from both (*n* = 6). The median of all ASP values was 12.1% (IQR, 6.8–20.3%; range, 1.1–43.5%). In lesions with ASP <12.9%, the median ASP was 6.9% (IQR, 3.5–8.5%; range, 1.1–11.9%) and significantly lower (Mann–Whitney U test, *p* < 0.001) compared to lesions with ASP >12.9% (median, 20.0%; IQR, 16.3–26.7%; range, 14.0–43.5%). Routinely acquired clinical characteristics are illustrated in Table 1.

### 3.2. Progression-Free Survival Analysis

The median PFS of the total cohort of 30 patients was 11.5 months (IQR, 6.0–17.6 months; range, 3.0–46.2 months; Figure 2). During follow-up in an everolimus-based regime, all 30 patients showed progressive disease (recurrence of the primary tumor or progressive tumor manifestations). New lesions were detected in 26/30 (87%) patients; new liver lesions were identified in 16/30 (53%), lymph node metastases in 5/30 (17%), bone metastases in 4/30 (13%) patients and lung metastases in one (3%) patient.

### 3.3. Cox Regression for PFS

Primary tumor site (pancreas vs. small bowel), ASP, number of metastatic sites and presence of bone metastases were significant predictors of PFS in univariable Cox regression (Table 2).

Log-rank test for ASP for PFS showed significant results at any cut-off between 8% and 32% (Figure 3); the optimal cut-off was >12.9% (Figure 4). In multivariable Cox regression, increased ASP >12.9% remained a significant predictor of shortened PFS while prior treatment with PRRT significantly predicted longer PFS (Table 3). The two variables “presence of bone metastases” and “number of metastatic sites ≥2” were mostly redundant (25 of 30 cases concordant; McNemar’s test, *p* = 0.063). Therefore, based on the lower *p* value in univariable analysis, only “presence of bone metastases” was included in multivariable analysis.

### 3.4. Prediction of PFS

Kaplan–Meier analysis revealed a median PFS of 6.7 months in patients with high ASP >12.9% (95% CI, 2.1–11.4 months) compared to 14.4 months in patients with low ASP ≤ 12.9% (95% CI, 12.5–16.3 months; log-rank test, *p* = 0.028; Figure 5).

## 4. Discussion

This retrospective bicentric study investigated the prognostic value of lesional asphericity (ASP) in pretherapeutic [^111^In-DTPA^0^]octreotide (OctreoScan) SPECT/CT in metastatic GEP-NET prior to everolimus treatment. High ASP predicted PFS independently from common clinical and histological parameters. The median Ki-67 index 5% represented well- and intermediate differentiated GEP-NET, and median PFS of all patients was 11.5 months. The median PFS was similar to that published in the RADIANT-3 trial with 11 months in P-NET [4] as well as RADIANT-4 with again 11 months PFS in advanced, progressive, and nonfunctional GI- and lung-NET [5]. In our study, P-, GI-, and CUP-NET were included irrespective of prior ”first-line” treatment strategies, and therapy was combined with somatostatin analogs in most of the cases. Hence, our PFS is comparable with a number of subgroup analyses of the RADIANT-2 trial, which supports the efficacy of everolimus in combination with long-acting repeatable octreotide (octreotide LAR) with a PFS of 13.6 months [3].

Several previous studies investigated different biomarkers and immunohistochemical parameters aimed at differentiation of responders and non-responders to treatment with everolimus but mostly failed [17,18]. Yao et al. reported shorter PFS and overall survival in patients with P-NET with elevated baseline CgA or neuron-specific enolase (NSE) [19]. In a study by Benslama et al. [20], hypercholesterolemia was associated with longer PFS, whereas the presence of bone metastases and overexpression of phosphorylated p70S6 kinase (p-p70S6K) by tumor cells were associated with shorter PFS under everolimus. However, the authors concluded that the absence of bone metastases and high levels of p-p70S6K are unspecific surrogates independently of therapy regime. Moreover, similarly to the histological tumor grade, they are generally known to show a strong association with a worse outcome.

It has, however, been demonstrated that there are different interactions and synergies between the pathways for SSR and mTOR inhibitors. In particular, several preclinical studies have recently shown that everolimus in conjunction with somatostatin might be acting in synergy on common signaling cascades, such as the PI3K-AKT-mTOR pathway [21]. Another explanation for synergistic effects might be the complementary effect of everolimus and somatostatin analogs on the protein 4E-BP1—a cell-signaling hallmark in NET [22,23]. Those synergies potentiate the inhibitory effect on tumor growth, reduce tumor perfusion, and release endogenous immunomodulation, which directly effects antitumor mechanisms.

It is well known that octreotide, a synthetic somatostatin analog, preferentially binds to the SSR subtype 2 and 5. There have been five SSR subtypes (SSTR 1–5) identified, and they exert direct antitumor effects. The prognostic importance of ASP derived from SSR imaging has been recently demonstrated in pretherapeutic and intratherapeutic SPECT/CT imaging in patients with NET undergoing [^177^Lu]Lu-DOTATOC-PRRT [9,10]. Thus, it may be hypothesized that neuroendocrine metastases exhibiting a homogeneous and uniform SSR expression could also respond especially well to everolimus and a concomitant treatment with SSA. High ASP values of SSR-positive lesions represent high tumor heterogeneity, thus hinting at dedifferentiated tumor cells and especially, necrosis (Figure 1). Finally, necrosis is known to be accompanied by disruption of the outer cell membrane integrity, initiating inflammation and edema [24]. Therefore, progressive necrosis causes pruning of tumor vessels that correlates with a significant reduction in locoregional blood flow. Poor perfusion may directly impair the therapeutic efficacy of blood flow-dependent treatments, such as everolimus.

Applying the ASP in pre- and intratherapeutic octreotide scintigraphy, it was recently demonstrated that lesions with low ASP (optimal cut-off <5.5% showed the best response to PRRT with [^177^Lu]Lu-DOTA-TATE) [10]. Lesions with homogenous SSR distribution and a regularly shaped SSR-positive lesion volume may facilitate a uniformly high lesional irradiation dose. The optimal cut-off value of 12.9% identified in the current patients was higher. While these two studies may only be compared very cautiously, the combined results might indicate that patients with ASP values between 5.5% and 12.9% might be candidates for treatment with mTor-inhibitors rather than PRRT. According to the North American Neuroendocrine Tumor Society (NANETS) guidelines for the treatment of advanced midgut NETs from 2017 [25], everolimus is also recommended as first-line therapy in patients who are no candidates for SSAs (e.g., weak or absent SSR expression). It is important to notice that the definition ”weak“ is a challenging definition for interdisciplinary consensus. So far, only the Krenning score is used to qualitatively describe the avidity of SSR-positive lesions [12]. If the current results can be confirmed, lesional ASP in SSR imaging may facilitate decision making for or against everolimus.

The current study is limited by its retrospective nature and lack of a matching control group undergoing different treatment. Ultimately, the conclusion that ASP not only shows a prognostic value but also offers a predictive role and can directly guide treatment decisions in patients eligible for several treatment options including everolimus would require comparative investigation of the predictive value of ASP in matched cohorts of NET patients undergoing these different treatments. Further limitations may arise: the cumulative dose (CD) and dose intensity (DI) of everolimus per patient were not assessed. Berardi et al. reported that both CD and DI play a prognostic role for patients with advanced P-NET [26]. A high number of patients had to be excluded from the current analysis because they suffered from everolimus-induced intolerable toxicity or required any dose reduction.

As a further limitation, in the current analysis, eight patients underwent liver-directed treatments including treatment of solitary liver metastases and uni-/bilobar selective internal radiation therapy (SIRT). The presence of pretreated metastases might introduce a bias as these lesions could show especially high ASP due to treatment-induced cellular necrosis, hypoxia-induced angiogenesis, and proliferation. Therefore, in these patients, lesions that had been previously treated with microwave or radiofrequency ablation were excluded from the analysis. Consequently, only two patients with prior SIRT were included, and we believe that this small number does not have a substantial impact on the main results. Mainly due to the remarkable progress and the rising clinical impact of SSR imaging [27], the concept of the ASP will be even transferred to functional PET imaging when enrolling prospective studies. Such prospective studies are required to validate the current explorative results and to ensure a well-selected, homogenous patient collective. Ideally, future studies should use SSR-specific PET/CT instead of SPECT/CT imaging, because the substantially better spatial resolution of PET would allow a more differentiated reflection of the lesions’ irregularity, especially in smaller lesions. Consequently, the currently identified ASP cut-off would not be directly transferable to PET imaging in the same context.

## 5. Conclusion

High lesional ASP in pretherapeutic SSR imaging as well as no prior performed PRRT independently predicted shorter PFS in patients with metastatic SSR-positive GEP-NET undergoing treatment with everolimus. Considering a broad spectrum of therapeutic strategies in well-differentiated NET, ASP could help to identify patients who are less likely to show favorable PFS with everolimus. Prospective, comparative investigation of patients undergoing different second- or third-line treatments is required to validate the prognostic and predictive value of ASP in this clinical setting.

## Figures and Tables

**Figure 1 diagnostics-10-00732-f001:**
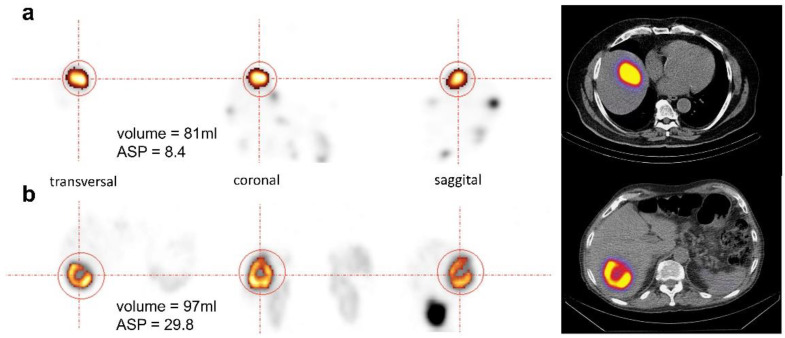
Orthogonal images of two representative examples of neuroendocrine tumors (NET) liver metastases with comparable metabolic tumor volumes (MTV)s, 81 mL (**a**) and 97 mL (**b**), but different asphericity (ASP) values: 8.4% (**a**) and 29.8% (**b**). The delineated somatostatin receptor (SSR)-positive MTV is highlighted by colouring.

**Figure 2 diagnostics-10-00732-f002:**
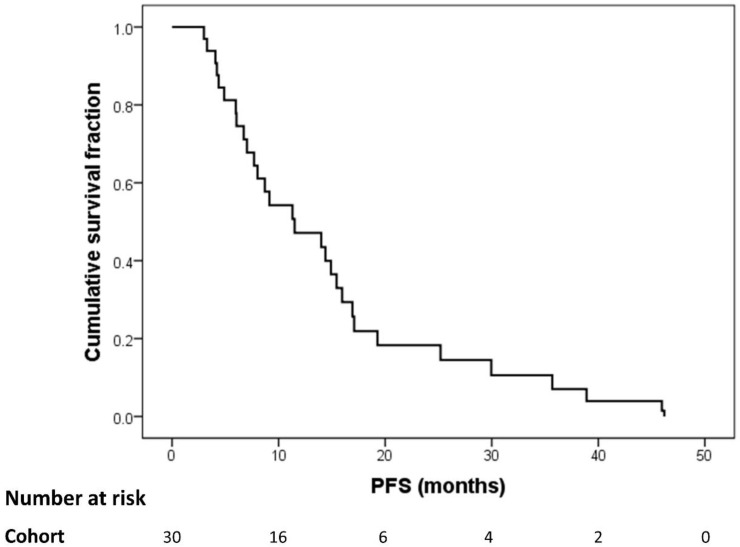
Kaplan–Meier curve for progression-free survival (PFS) with everolimus for the total cohort.

**Figure 3 diagnostics-10-00732-f003:**
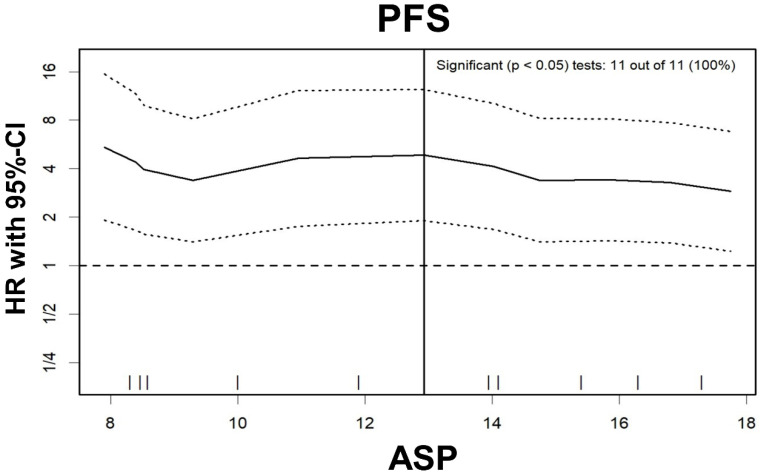
Hazard ratio plot for asphericity (ASP). The hazard ratio (HR; solid line) and its 95%-confidence interval (dashed lines) are illustrated for ASP cut-offs between 8–18%. A dashed horizontal line represents equal PFS in patients with high or low ASP, respectively (i.e., HR of 1.0). The vertical solid line displays the optimal ASP cut-off (12.9%). Furthermore, the frequency of possible ASP cut-off values between 8–18% with significant results in the log-rank test is given (11 out of 11 possible cut-offs).

**Figure 4 diagnostics-10-00732-f004:**
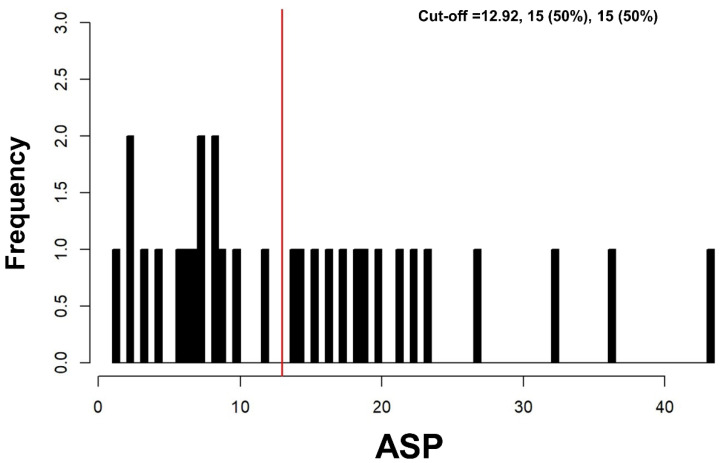
Histogram—Cut-off finder for asphericity (ASP). Histogram showing the distribution of ASP among all 30 patients. The vertical line (red) highlights the optimal cut-off at 12.9%.

**Figure 5 diagnostics-10-00732-f005:**
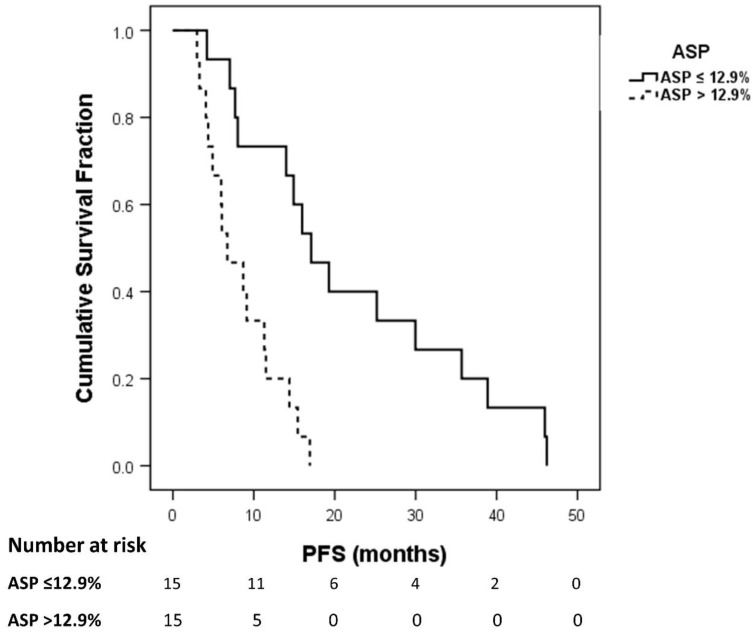
Kaplan–Meier plot of progression-free survival (PFS) for patients with high versus low asphericity (ASP; optimal cut-off, >12.9%).

**Table 1 diagnostics-10-00732-t001:** Patient data and characteristics.

Patient Characteristics	Variable	Value (%)
Total		30 (100)
Sex	female male	13/30 (43) 17/30 (57)
Age in years	median (range)	67 (48–81)
Primary		
	pancreatic	17/30 (57)
	duodenum/ileum/jejunum	9/30 (30)
	CUP-NET	4/30 (13)
Previous treatment	surgery of the primary	19/30 (63)
	Liver-directed treatment ^a^	8/30 (27)
	SSA	27/30 (90)
	PRRT	6/30 (20)
	chemotherapy	7/30 (23)
Number of evaluated liver metastases per patient		
	*n* = 1 (% of patients)	3 (10)
	*n* = 2 (% of patients)	10 (33)
	*n* = 3 (% of patients)	17 (57)
Administered activity, MBq (OctreoScan^®^)	Median (range)	222 (181.3–265.5)
Ki-67 (%)	Median (range)	5 (1–25)
Grade		
	G_1_	9/30 (30)
	G_2_	19/30 (63)
	G_3_	2/30 (7)
Chromogranin A (µg/L)	Median (range)	482 (47–45,800)
ASP (%)	Median (range)	12.1 (1.1–43.5)
	ASP (%) < 12.9	6.9 (1.1–11.9)
	ASP (%) > 12.9	20.0 (14.0–43.5)

Detailed patient characteristics in systematic overview: CUP-NET, neuroendocrine neoplasm of unknown primary; ASP, asphericity; SSA, somatostatin analogs; PRRT, peptide receptor radionuclide therapy; a = debulking surgery or minimal invasive techniques (e.g., selective radiofrequency ablation).

**Table 2 diagnostics-10-00732-t002:** Univariable Cox regression for progression free survival (PFS).

Variable	HR	95%-CI	*p* Value
**Patient Variables**			
Male sex	2.03	0.91 to 4.5	0.082
Age in years (continuous)	1.0	0.96 to 1.04	0.93
**Primary Tumor Site**			
Pancreas		Reference	0.084
Small bowel	0.34	0.13 to 0.89	**0.028**
CUP	0.46	0.16 to 1.35	0.158
**Biological Variables**			
Baseline CgA ≥ 2 ULN	0.9	0.38 to 2.14	0.811
Ki-67 in % (continuous)	0.98	0.92 to 1.05	0.575
ASP in % (continuous)	1.06	1.03 to 1.10	**0.001**
**Ki-67 Grading**			
G_1_		Reference	0.327
G_2_	1.06	0.48 to 2.33	0.887
G_3_	3.22	0.67 to 15.5	0.144
**Clinical Parameters**			
Bone metastases	3.72	1.57 to 8.83	**0.003**
Lymph node metastases	1.29	0.54 to 3.05	0.568
Number of metastatic sites ≥ 2	2.37	1.09 to 5.15	**0.03**
**Prior Treatment**			
Primary tumor surgery	1.64	0.70 to 3.82	0.251
Liver-directed treatment	0.79	0.65 to 3.19	0.579
SSA	2.18	1.08 to 7.35	0.21
PRRT	0.32	0.12 to 0.87	**0.025**
Chemotherapy	1.86	0.77 to 4.48	0.167

Significant results are printed in bold. ULN, upper limit of normal. HR, hazard ratio; 95% CI, 95%-confidence interval. CUP, neuroendocrine neoplasm of unknown primary. SSA, somatostatin analogs. PRRT, peptide receptor radionuclide therapy.

**Table 3 diagnostics-10-00732-t003:** Multivariable Cox regression for progression-free survival (PFS).

Variable	HR	95%-CI	*p* Value
ASP (>12.9 vs. ≤12.9%)	3.33	1.17 to 9.47	**0.024**
Bone metastases	1.89	0.67 to 5.34	0.231
Prior PRRT	0.35	0.12 to 0.97	**0.043**
Primary tumor site			
Pancreas		Reference	0.855
Small bowel	0.79	0.27 to 2.30	0.664
CUP	1.03	0.34 to 3.12	0.954

Significant results are printed in bold. HR, hazard ratio; 95%-CI, 95%-confidence interval. ASP, asphericity; PRRT, peptide receptor radionuclide therapy; CUP, neuroendocrine neoplasm of unknown primary.

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
