# Peer review of "Asphericity of Somatostatin Receptor Expression in Neuroendocrine Tumors: An Innovative Predictor of Outcome in Everolimus Treatment?"

_diagnostics, 2020, doi:10.3390/diagnostics10090732_

Round 1

Reviewer 1 Report

The present manuscript is a retrospective evaluation on 30 patients with well-differentiated NET undergoing Everolimus therapy. The authors analyze the impact of clinical factors and SPECT derived textural features – namely asphericity – on PFS. They identified a cut-off of 12.9% for prediction of PFS (alongside prior PRRT) in multivariate analysis. Asphericity seems to represent a major topic of the work group as several manuscripts have been published for that matter. The number of patients is relatively low, however, the manuscript is well written, the data is well described and interpreted, and will be of interest to the NET community. Nonetheless there are some aspects that need further clarification and discussion.

Abstract:

No comments.

Introduction:

Well written, informative and good introduction to the aim of the study.

Methods:

  • Please define “substantial dose reduction” for everolimus. Many patients need dose reduction by 50% (which is substantial). What was the dose limit for patient exclusion? How many patients have been excluded? Also relevant for the last paragraph of the discussion.

Results:

  • Please provide more information on the lesions where asphericity was obtained. Was it only liver metastases? This is highly important considering that 11 patients had liver directed therapy. The form of treatment (e.g. TACE, SIRT, RFA) might induce focal necrosis and thus impact asphericity. Were there any differences in ASP between patients with and without prior liver directed therapy? Please add this to the discussion, too.
  • Table 2: What were the different grades described for asphericity?
  • What was the median ASP?
  • Please just provide median PFS and omit mean PFS. And which values are correct (15.1 in the results, 11.5 in the discussion on page 8)

Discussion:

  • Why did the authors not discuss their results with their own findings for ASP cut-offs in PRRT? This is particularly interesting as a) the current workgroup is one of the main groups focusing on ASP and discussion of results is rare and b) the cut-off values are different compared to PRRT (e.g. 5.12 in [10] and 5.5 in [11]).
  • How do the authors conclude that patients with ASP between 5.1 and 12.9 will rather from everolimus than from PRRT. Their data do not analyze this in the current form (e.g. discuss with cut-off values mentioned above).
  • The authors should add a comment on whether their data is also expandable to PET imaging (which we all assume, of course) as it represents the reference standard for NET Imaging.

Minor:

  • Please insert citation for “Krenning-Score”
  • Results, Patients: there seems to be a change in the font

Reviewer 2 Report

This study was conducted to examine whether ASP of NET lesions in SSR imaging before initiation of everolimus treatment can predict PFS. The authors concluded that pretherapeutic ASP of SSR positive lesions predicted PFS for treatment with everolimus in GEP-NET. The image-based strategy of outcome prediction before therapy might be useful for patient risk stratification. However, there were several questions and comments.

  1. Statistical analysis: Multivariable Cox regression was performed, including significant parameters in univariable Cos regression (Line135-136).->Why did authors exclude "number of metastatic sites" from multivariate cox regression analysis?
  2. Patients. Is it appropriate to include G3 patients? 
  3. Abstract, Table1: The average age of patients is different between abstract (median age, 66.5 [48-81] years) and table [median age, 59 (38-75)].
  4. Patients: The study protocol was approved by the Otto-von-Guericke-University Institutional Committee in accordance with the ethical standards of the institutional research committee (reference ID: RAD318; vote, 10/2016) and with the 1964 Helsinki declaration and its later amendments or comparable ethical standards.->This study is a bicentric cohort study. Did another institution approve the study?
  5. SPECT/CT imaging: Acquisition parameters are missing. The parameters are elements for analysis results.
  6. Image analysis: For lesion-based analysis, a VOI was delineated separately for each liver metastasis... on visual assessment were delineated. ->How was the inter-observer variability with assessment?
  7. Image-based response evaluation: Lesions with a short axis of <10 mm (e.g., lymph nodes) were excluded from the analysis to avoid partial volume effects.-> Is this description correct? 
  8. Figure1. Please add the image orientation.
  9. Figure2 and 5. The number at risk are missing.
  10. There was no mention of additional treatment after everolimus.

Round 2

Reviewer 2 Report

The authors responded to all comments and revised the manuscript sufficiently.

However, I would like to confirm one thing.

Fig.5 The cumulative survival fraction

Is 0 of the number at risk in the "ASP >12.9%" group at 20 months correct?
Maybe this curve concludes the censored case. Please check it out.